# Flexural Fatigue in a Polymer Matrix Composite Material Reinforced with Continuous Kevlar Fibers Fabricated by Additive Manufacturing

**DOI:** 10.3390/polym14173586

**Published:** 2022-08-30

**Authors:** Alberto David Pertuz-Comas, Jorge G. Díaz, Oscar Javier Meneses-Duran, Nixon Yesid Niño-Álvarez, Juan León-Becerra

**Affiliations:** 1GIC, Escuela de Ingeniería Mecánica, Universidad Industrial de Santander, Carrera 27 Calle 9, Bucaramanga 680002, Colombia; 2GIEMA, Escuela de Ingeniería Mecánica, Universidad Industrial de Santander, Carrera 27 Calle 9, Bucaramanga 680002, Colombia; 3Escuela de Ingeniería Mecánica, Universidad Industrial de Santander, Carrera 27 Calle 9, Bucaramanga 680002, Colombia

**Keywords:** Additive Manufacturing, fatigue, finite element method, bending stress

## Abstract

Fatigue bending tests, under controlled displacement, were performed on a polymer matrix composite material reinforced with continuous Kevlar fibers. The samples were fabricated using the Fused Filament Fabrication (FFF) technique in a Markforged Two^®^ 3D printer. The static characterization delivered a flexural modulus of elasticity of 4.73 GPa and flexural strength of 110 MPa. The applied loading corresponded to 92.3, 88.5, 86.2, and 84.7% of the static flexural displacement, giving 15, 248, 460, and 711 cycles for failure. Additionally, two numerical models were created: one using orthotropic properties for static loading conditions; and a second one using isotropic in-bulk properties for fatigue modeling. The second model was able to reproduce the experimental fatigue results. Finally, morphological analysis of the fractured surface revealed fiber breakage, fiber tearing, fiber buckling, matrix cracking, and matrix porosity.

## 1. Introduction

Additive Manufacturing (AM) enables the production of geometrically complex parts without post-processing. AM is used in a wide range of materials and allows the reconstruction of layer-by-layer 3D topologies that, with traditional manufacturing processes such as milling, cutting, casting, forging, and welding, would be complicated to manufacture [1,2,3,4,5]. AM could effectively link topological optimization and the final product by delivering final geometries with no additional processing [2]. Furthermore, it has been estimated that printing costs are independent of the batch size [6]. Besides, being such a recent technology, AM is a technique in constant development [2], which has been integrated into manufacturing chains [7]. FFF is by far the most widely used method of AM techniques [2,4,5,7]. The AM technology provided by Markforged^®^ (Markforged, Watertown, MA, USA) [8] can provide load-bearing parts matching the strength of aluminum [9,10]. 

On the other hand, fatigue is a common failure mechanism responsible for 90% of rotating equipment failures in the industry [11]. Methods for characterizing materials under fluctuating loads include stress versus life (*σ–N*) for loads below the elastic limit, strain versus life (*Ɛ**–N*) for loads above the elastic limit but below the maximum stress that produce crack nucleation, and the *da/dN* versus *ΔK* method when measurable cracks are present, using Paris-type rules to estimate crack growth [11]. In composites, the interlaminar fracture toughness is usually performed [12]. Saleh et al. [13] stated that a composite material is considered to fail when its residual strength is less than 85% of its ultimate tensile stress (UTS). Therefore, it is imperative to establish relevant mechanical properties to design components that withstand appropriate service conditions [3,11]. Recent works [14,15,16] presented detailed studies and experimental data under uniaxial static and fatigue loads for composite materials made of a Nylon matrix (PA 6), reinforced with continuous fiber and printed by FFF for different filling densities, filling types, and fibers. In [17,18,19] FFF materials made of an Onyx^®^ matrix with no continuous fiber reinforcements were studied and characterized. Onyx^®^ is, in turn, a composite of a polymeric matrix made out of Nylon with short carbon fiber reinforcements [19]. Because this polymeric matrix is new, there are not many testing data on it yet. Moreover, Nylon is known to be a time-dependent material; UV and water affect its mechanical properties [11]. Recently, it was shown that printing direction might affect relaxation on other viscoelastic materials as well [20]. Therefore, a model that accounts for both may be needed. However, the static behavior has been shown as linear elastic for Nylon [9,21,22] and PLA [6,23] matrices.

Recent reviews on the state of the art, current trends, and limitations, and FFF and AM, are found in [1,24,25,26,27,28], whereas some reviews on tests and experimental techniques for different failure modes in FFF composites are found in [22,29,30]. Finally, Diaz et al. [9] identified some missing tests to understand the mechanical behavior of this class of FFF composite materials. This work presents the fatigue characterization under alternative four-point bending of a polymeric matrix composite material reinforced with embedded short fiber and continuous Kevlar fiber, manufactured by the novel FFF technique. We included numerical modeling and a microscopy analysis to explain the failure modes’ origin.

## 2. Background

This section gives the baseline knowledge needed to understand the paper.

### 2.1. Additive Manufacturing

AM is a technology created by Hull in 1984 that began with stereolithography, followed by FFF in 1989 [5]. After the Stratasys patent expired in 2010, there was a spike in FFF development. In 2015, Markforged^®^ (Watertown, MA, USA) [8] obtained a patent to manufacture FFF-printed composite materials reinforced with continuous fibers, giving good dimensional accuracy and stability [7].

FFF is a manufacturing technique in which topology is reconstructed layer by layer from a G-code file that has previously been converted to STL format. [4,5]. This process is done by software in the cloud called Eiger^®^ for Markforged^®^ composites [8]. The 1.75 mm polymeric filament, Onyx^®^ in this case, is fed into an extruder, heated above its melting point (260 °C, approx.), and deposited on a platform where it cools to form the wanted part. The extruder movement is controlled by software that optimizes the movements and alternates matrix and continuous reinforcement deposition. A schematic of the FFF printing process is shown in Figure 1.

Before printing a part, several process parameters must be selected in the Eiger^®^ (Watertown, MA, USA) web-based software [2]. These are fiber type (carbon, Kevlar, fiberglass, and high-temperature fiberglass), fiber volume fraction, fiber layout type (concentric and isotropic, as shown in Figure 2), matrix fill pattern (rectangular, hexagonal, triangular, as seen in Figure 2, and solid fill), matrix fill density, matrix deposition angle, and fiber deposition angle [15]. In addition, there are new developments in cellular structures that might give improved strength and stiffness over existing ones [31].

In the same way, as in the design with traditional composite materials, the properties of the parts produced by AM depend on the volumetric fiber fraction, the arrangement and angle of the fibers, and the mechanical properties of the reinforcement and matrix [28,29]. Additionally, FFF creates an anisotropy besides what classic composites show [32]. Therefore, it is common to represent its mechanical behavior with an orthotropic model [2] from matrix and fiber properties such as the ones provided by Markforged^®^ [33], as shown in Table 1. Melenka et al. [34] proposed a model to estimate mechanical properties in AM printed composite materials based on the volumetric average stiffness (VAS) method [35] but accounting for voids in the matrix [36]. 

As a comparison for the values presented in Table 1, [19] reported 7.8 MPa for average ultimate tensile strength for a triangular filling at 20%. An analytical summary of reported mechanical properties for a wide combination of parameters is available at [9]. Further details on the printing process may be found in [8].

### 2.2. Fatigue 

It is commonly accepted that fatigue failure begins with the nucleation of cracks [11]. For the most part, cracks in composites form in layers perpendicular to the load direction [9,26]; this process is called transverse matrix cracking and involves numerous microcracks. Thus, cracks in the matrix are generally the first form of damage in composites and function as a barrier delaying further macroscopic damage [30].

When a structural component is subjected to a repetitive load well below the yield stress, the recommended design method is stress life (*σ–N*). This behavior is described by the phenomenological Basquin rule [11], as shown in Equation (1).
(1)$$\sigma =AN{\;}^{b}\;$$
where *σ* is stress, the strength reduction rate depends on the number of cycles and material-dependent constants. A straight line is obtained when plotting Equation (1) in a bilogartimic graph (*N, σ*). However, the tests were done at a deflection inversion ratio, *Ry* = −1, as shown in Equation (2).
(2)$$R_{y}=\frac{Y_{\min}}{Y_{Max}}$$
where *Y* is the beam’s deflection. 

Two studies were found regarding fatigue studies about composite materials printed by Nylon matrix FFF [14,15]; both were done under axial load. This study takes the configuration that showed the best results when combined with different printing parameters [15,16], see Table 2. In [15], the experimental data, which was performed under a load inversion ratio R = −1, was fitted into Basquin’s rule [11]. Furthermore, these materials have been shown to exhibit a loss of stiffness under cyclic loads [27,30].

## 3. Materials and Methods

The specimens were manufactured according to ASTM D6272-17 (Standard Test Method for Flexural Properties of Unreinforced and Reinforced Plastics and Electrical Materials by Four-Point Bending) for static ASTM D7774-17 (Standard Test Method for Flexural Fatigue Properties of Plastics) for four-point bending fatigue tests. All were printed on a Markforged 2.0 printer using parameters defined in [15,16] with Onyx as a matrix with a 19% of continuous Kevlar fiber as reinforcement, both provided by Markforged. Table 2 shows the parameters used. Finally, an MTS Bionix^®^ (Minneapolis, MN, USA) servohydraulic 370.02 universal testing machine (UTS) was used for the static and fatigue tests at room temperature. The machine was equipped with a 25 kN load cell.

### 3.1. Static Test

Tension tests were performed using an MTS 634.12f 25 mm (Minneapolis, MN, USA) axial extensometer, as shown in Figure 3a, at a crosshead speed of 0.5 mm/min. Specimen dimensions were chosen according to ASTM D6272-17: 127 mm long, 12.7 mm wide, and 3.2 mm thick. For these tests, end-tabs [37] were used to guarantee the clamp-specimen grip, to reduce the compression stress due to an excessive grip force and, therefore, an unwanted failure due to stress concentration. The static bending tests were made at 6 mm/min crosshead speed, according to ASTM D6272. A dial gauge indicator, shown in Figure 3b, was used in the static bending test to verify the sample’s deflection. Finally, the in-house built bidirectional supports are shown in Figure 3c. 

The experimental stress was obtained according to a relationship recommended in ASTM D6272-17 and shown in Equation (3).
(3)$$\sigma =\frac{Px}{bd^{2}}$$
where *σ* is tension, *P* is force, *x* is the distance between the lower support and the point of application of the load, *b* is the width of the specimen, and *d* is the thickness of the specimen. The ratio for strain, ε, also recommended in ASTM D6272-17, is shown in Equation (4).
(4)$$\varepsilon =\frac{4.7dY_{\max}}{x^{2}}$$
where *Y_max_* is the maximum deflection of the beam.

### 3.2. Fatigue Tests

The alternative four-point fatigue tests were done at ambient temperature, under a displacement inversion factor *Ry = −1* under controlled displacement applying a 5 Hz sinusoidal load. An example of the control and feedback displacement signals is seen in Figure 4, taken from the MTS suite^®^ (Minneapolis, MN, USA) control software. For these tests, in-house supports were built to support loads in opposite directions with a 12 mm rod radius of, as shown in Figure 3c. The applied load levels corresponded to 92.3, 88, 86.2, and 84.7% of the maximum deflection obtained from the static tests. Furthermore, Castro and Meggiolaro [11] explained the effect on loading frequency; at higher frequencies fatigue life is shortened by the temperature rise that the polymer matrix is not capable of properly dissipating, most likely due to local cyclic heating. However, that effect is only significant when operating frequencies experiment a magnitude change of 10 or more [38]. Therefore, the results presented here must only be applied within the tested range.

### 3.3. Numerical Modeling 

The elastic problem for a composite material can be described using a linear elastic model and the Galerkin formulation to solve the displacements *u* through the domain Ω, as shown in Equation (5).
(5)$$\int_{\Omega }\varepsilon {\left(v\right)}^{T}E\varepsilon (u)d\Omega =\int_{\Omega }v^{T}bd\Omega +\int_{\Gamma }v^{T}td\Gamma$$
where *Ɛ* and *E* are strain and elastic tensors, respectively, *b* the body forces, and *t* the traction vector.

A finite element simulation was carried out on Ansys^®^ (ANSYS, Canonsburg, PA, USA). Two numerical models were created. The first model used detailed geometry using the ACP (Ansys Composite Module) with orthotropic properties to simulate the static bending test. Figure 5a shows the cross-section of the detailed model as arranged in the ACP model. Figure 5b shows the organization of the different layers of the specimen, and Figure 5c shows the orientation of the triangular fill layers. The second simulation was a simplified model using bulk properties, as suggested in [9]. The simplified model was used to simulate the fatigue tests because fatigue properties were unavailable for each material region. 

The properties of the cross-sectional regions for the detailed model were estimated through the Ansys Material Designer plug-in for Ansys Workbench, which is based on a homogenization method through an RVE (Representative Elementary Volume) model. These properties were then verified with the ROM and the Halpin-Tsai equations [39]. The triangular filling region in the simulation was considered a solid material but with equivalent properties calculated based on the equations of cellular solids, see Section 4.2 [35]. For the Onyx^®^ solid regions (layers and wall), the mechanical properties assigned to the numerical model were obtained based on the model described in [36] and shown in Table 3.

For the fatigue simulation in Ansys^®^ using the simplified model, the Onyx reinforced with continuous Kevlar fiber specimen was considered a solid beam with orthotropic properties. These properties were estimated analytically through the volume average stiffness method (VAS) proposed in [34]. Table 4 shows the values used.

### 3.4. Microscopy

Micrographs were taken with a scanning electron microscope (SEM) to observe the failed surface. First, samples were plated with gold and then placed into a Vega 3 Tescan SEM working between 5 and 10 keV equipped with a tungsten filament. The gold plating was done in some samples, finding no difference under the SEM with the no plated samples. The need for no plating was attributed to the chopped carbon fiber in the PA6 matrix that made the sample conductive.

## 4. Mechanical Properties Estimation

This section details how the mechanical properties used in the numerical models were estimated.

### 4.1. Triangular Fill Properties Estimation 

Gibson and Ashby [35] proposed the concept of relative density, *p_r_* shown in Equation (6), based on fill density, *p_x_*, and fill material density *p_s_*, for a triangular fill.
(6)$$p_{r}=\frac{p_{x}}{p_{s}}$$

For equilateral triangle fills *p_r_* is given by Equation (7), where *t* is the width of the triangular cell, and *l* is the length of the triangular cell.
(7)$$\begin{array}{l} E_{1}=E_{2}=1.15E_{s}\frac{t}{l}\\ E_{3}=E_{s}p_{r}\\ \rho_{Tr}=\frac{2\sqrt{3}t}{l} \end{array}$$

The shear modulus in the different planes (*G*_12_, *G*_23_, *G*_13_) were calculated with the equations shown in Equation (8) based on *Gs*, Onyx shear module.
(8)$$\begin{array}{l} G_{12}=0.125G_{s}p_{r}\\ G_{13}=G_{s3}=0.5G_{s}p_{r}\\ G_{s}=\frac{E_{s}}{2(1+\upsilon_{s})} \end{array}$$

Finally, the Poisson ratio in the different planes (*v*_12_, *v*_23_, *v*_13_) was estimated by the Equation (9), where *v_s_* is Onyx Poisson ratio.
(9)$$\begin{array}{l} \upsilon_{12}=0.333\\ \upsilon_{23}=\frac{E_{2}}{E_{3}}\upsilon_{s}\\ \upsilon_{13}=\frac{E_{1}}{E_{3}}\upsilon_{s}\\ \upsilon_{s}=0.35 \end{array}$$

### 4.2. Solid Onyx Region Properties Estimation 

The upper and lower walls and layers were characterized following Rodríguez [31], which considered the level of porosity (*p*_1_) of the solid regions printed by FFF. Thus, the material is deposited in the Z direction, as shown in Figure 6.

The elastic modulus in the longitudinal direction (*E*_1_) and in the transversal directions (*E*_2_, *E*_3_) are given by Equation (10), which is modified by the porosity level *p*_1_. According to Papon and Haque [40] *p*_1_ is estimated at 10%.
(10)$$\begin{array}{l} E_{1}=\left(1-p_{1}\right)E\\ E_{2}=\left(1-\sqrt{p_{1}}\right)E\\ E_{3}=E_{2} \end{array}$$

For calculating the shear modulus in the different planes (*G*_12_, *G*_23_, *G*_13_), Equation (11) was used.
(11)$$\begin{array}{l} G_{12}=G\frac{\left(1-p_{1}\right)\left(1-\sqrt{p_{1}}\right)}{\left(1-p_{1}\right)+\left(1-\sqrt{p_{1}}\right)}\\ G_{23}=G_{13}=\left(1-p_{1}\right)G \end{array}$$

Finally, for calculating the Poisson’s ratios in the different planes (*v*_12_, *v*_23_, *v*_13_), Equation (12) was used.
(12)$$\upsilon_{23}=\upsilon_{13}=\upsilon_{12}=\upsilon \left(1-\sqrt{p_{1}}\right)$$

### 4.3. Estimation of the Properties for the Continuous Fiber-Reinforced Regions

The continuous Kevlar fiber-reinforced regions, see Figure 5a, were considered a unidirectional ply composed of a Nylon matrix and continuous fiber, with equivalent properties based on the rule of mixtures (ROM) and the Halpin-Tsai equations [39].

The modulus of elasticity in the longitudinal direction (*E*_1_) and the transverse direction (*E*_2_, *E*_3_) were calculated with Equation (13).
(13)E1=EfVf+(1−Vf)EmE2=E3=Em[1+nξVf1−nVf]n=EfEm−1EfEm+ξ
where *Vf* is fiber content, and *ξ* is 2 for circular fibers. The shear modulus in the different planes (*G*_12_, *G*_23_, *G*_13_) were calculated with Equation (14).
(14)$$\begin{array}{l} G_{12}=G_{m}\left[\frac{1+V_{f}+\frac{(1-V_{f})G_{m}}{G_{f}}}{1-V_{f}+\frac{(1+V_{f})G_{m}}{G_{f}}}\right]\\ G_{12}=G_{m}\left[\frac{V_{f}+(1-V_{f})n_{4}}{\left(1-V_{f}\right)n_{4}+\frac{V_{f}G_{m}}{G_{f}}}\right]\\ n_{4}=\frac{3-4\left(1-V_{f}\right)+\frac{G_{m}}{G_{f}}}{4V_{f}} \end{array}$$
where *Vf* is 0.36 and *v_m_* is 0.39. To calculate Poisson’s ratios (*v*_12_, *v*_23_, *v*_13_) Equation (15) was used.
(15)$$\begin{array}{l} \upsilon_{12}=\upsilon_{f}V_{f}+\left(1-V_{f}\right)\upsilon_{m}\\ \upsilon_{23}=\upsilon_{13}=\frac{k^{\prime }-mG_{23}}{k^{\prime }+mG_{23}}\\ m=1+4k^{\prime }\frac{\upsilon_{12}{}^{2}}{E_{1}}\\ k^{\prime }=\frac{\left[K_{m}\left(K_{f}+G_{m}\right)\left(1-V_{f}\right)\right]+\left[K_{m}\left(K_{m}+G_{m}\right)V_{f}\right]}{\left(K_{f}+G_{m}\right)\left(1-V_{f}\right)+\left(K_{m}+G_{m}\right)V_{f}}\\ K_{f}=\frac{E_{f}}{\left[2\left(1+\upsilon_{f}\right)\right]\left(K_{m}+G_{m}\right)V_{f}}\\ K_{m}=\frac{E_{m}}{\left[2\left(1+\upsilon_{m}\right)\right]\left(K_{m}+G_{m}\right)V_{f}} \end{array}$$

## 5. Results and Discussion

### 5.1. Static Tests

Figure 7 shows the results of the uniaxial tension test for the non-reinforced Onyx^®^ sample performed with an MTS 634.12F extensometer. The sample had the same printing parameters as the bending specimens, as the best performing ones reported in [15], but did not have continuous Kevlar fiber reinforcement. Stress was calculated as average stress; this is force over the initial area, while strain was estimated as elongation over initial length. Figure 7 depicts exemplary results for the tension test, where a 12.5 MPa maximum stress and 10.84% strain at rupture are seen. As a comparison, Barnik et al. [19] reported for solid Onyx a maximum stress of 31 MPa, whereas Markforged listed 40 MPa, see Table 1. For other AM polymeric materials, Parrado [6] reported a maximum of 48.4 ±1.9 MPa for solid PLA but for woven-layered (0 ± 90°) samples. Another source of difference could be attributed to the sample´s dimensions [9].

Moreover, an expression based on the Euler-Bernoulli beam theory showed that the maximum deflection is 1.15 times the deflection measured at the central supports, where the load is applied. The measured deflection was verified with a dial gauge. Figure 8 shows exemplary results of four points bending tests for three samples. An average modulus of elasticity of 4.37 ± 0.65 GPa and average ultimate stress of 109.9 ± 14.4 MPa were obtained.

### 5.2. Fatigue Tests

The tests were carried out under controlled displacement. The specimens were distributed in four displacement levels corresponding to 92.3%, 88.5%, and 86.2% of the maximum flexural displacement, corresponding to ±12, ±11.5, ±11.2, and ±11 mm, respectively. From these tests, the applied stress and number of failure cycles, *N_f_*, were obtained. Results of the tested specimens are shown in Table 5.

A drop in stiffness was observed in the load-displacement cycles, as reported by Saleh et al. [13]. It is understood with this behavior that stiffness changes because of the matrix collapse or the rupture of some of the reinforcing fibers. The energy absorbed by the composite material, represented by the shape of the hysteresis loop, seen in Figure 9, results in matrix-fiber separation, generating voids that act as stress concentrators, thus reducing fatigue resistance. This behavior is consistent with the literature where this failure mechanism has been mentioned [26,39]. Exemplary results of this phenomenon are shown in Figure 9, where one can observe that the load required to produce the ±11 mm constant deflection lowers every loading cycle. It is observed how the maximum force lowers from 170 N at the first cycle, goes to 146 N at ten cycles, 124 N at 20 cycles, and stays about 116 N from 100 all the way to 300 cycles. Moreover, the crack was not visible until beyond the 85% loss of strength predicted by Saleh et al. [13]. Therefore, we found that Saleh et al. [13] criteria is conservative as it predicts shorter lives as the sample experiences.

Moreover, the force-displacement loops do not seem to behave linearly as the loading and unloading paths do not follow the same route. From 1 to 2, the load raises as the deflection rises, but somewhere in between, the slope raises; the deflection causes a more rigid sample, so, the sample acts as a bending spring. At 2, the maximum load and displacement are reached. The displacement is inverted, and between 2 and 3, the load drops rapidly but at a lower slope than it did between 1 and 2. This may be due to the stored energy in the sample during the loading stroke, loading path 1 to 2. The load goes negative at 4, and the slope raises again between 4 and 5. It repeats the process seen between 1 and 2 but at negative loads. At 5, the sample reaches a peak for both displacement and load. The unloading occurs right after 5 but again at a different slope than between 4 and 5, perhaps in the same manner that it occurred for a positive load (from 2 to 3); the stored energy helps to bring back the sample to zero force. At 6, and very rapidly, the sample reaches almost zero force but is far from the zero-displacement position. Finally, the displacement reaches 1 to start another cycle. 

### 5.3. Numerical Simulation

A mesh size analysis was made to guarantee the convergence of the solution. The displacement is not dependent on mesh size, as seen in Figure 10. After the second mesh size, about 17,000 elements, corresponding to an average of 1 mm elements, the displacement remains about the same. With this mesh size, eight load values were analyzed from 0 to 70 N, as seen in Figure 11, where numerical results are plotted and compared with experimental tests.

Four load levels corresponding to 92.3, 88.5, 86.2, and 84.7% of the flexural displacement were simulated for the alternative fatigue results. The fatigue life of the material was compared between experimental results and the numerical model. The fatigue simulation used the simplified numerical model described in Section 3.3. The results are seen in Figure 12, where both plots (numerical and experimental) run parallel and very close to each other, with a low as 2.5% difference. 

Additionally, results from similar samples (Kevlar reinforced and triangular filling, but axially loaded instead of bending and PA 6 matrix instead of Onyx^®^) retrieved from the literature [15] are presented in Figure 12. In an axially loaded rectangular sample, the normal stress is assumed as average in the whole section, whereas in a bending sample, only half of the section is in tension and the other half in compression. However, in bending, the stress gradient depends on the distance from the neutral plane. The bending samples show lower stress than the axially loaded counterparts. Thus, besides the stress gradient, this stress difference may be attributed to the short fiber embedded in the Onyx^®^ matrix, present only in bending samples. The space left on the Onyx^®^ matrix for the chopped carbon fiber may act as a stress concentration factor lowering the fatigue endurance.

So, experimental data from Figure 12 were adjusted to Basquin’s rule [11] Equation (1), giving the results shown in Equation (16) that represent 95.2% of the experimental data. However, the simulation showed a trend to overpredict failure cycles for the upper part of the simulated cycles. For example, the observed cycles for the +/−11 mm were 711, whereas the simulation gave 755. On the other hand, the value of parameter *b* in Equation (16) was adjusted to −0.074, showing that it is indeed a brittle composite, as previously reported [15].
(16)$$\sigma =113.55N_{f}{\;}^{-0.074}\;$$

Finally, the difference between experimental and simulated results went from 33.3% at low cycles to 6.2% for the longest recorded cycles.

### 5.4. Failure Analysis

Figure 13 shows a general view of three fatigue surfaces taken by a digital camera. The macroscopic crack is nucleated in the polymer matrix on the most strained surface and propagates until it reaches the fiber layer. This nucleation can be attributed to the pure bending moment in the sample´s central zone, which produces normal stress. The combination of the remaining Onyx^®^ matrix, fibers, and nucleated surface cracks account for the elasticity module of the composite. The Kevlar fibers, which are stronger but more brittle, fracture first, starting with the closest to the external surface and consequently changing the composite stiffness. Thus, the elastic modulus changes with every fractured fiber. Such behavior explains the observed decrease in absorbed energy seen in Figure 9. Some samples presented wrinkles on the surface with no evident crack, see Figure 13. Such morphology may be attributed to fiber buckling [9,30]. The continuous fibers are slender, and during the reversal load, they experience compression, which may induce buckling, whereas the matrix has not failed yet. Such buckling may induce fiber-matrix separation. Therefore, composites may fail due to fatigue even under a negative load.

Samples were submerged in liquid nitrogen and fractured by a sudden load to observe the failed surface. Figure 14 shows layers of molded polymer (Onyx), with their respective aggregates of short carbon fiber distributed in the Nylon matrix; the distributed fibers appear to be stress concentrators. Empty spaces can be seen between each layer of polymer, and details of the porosity can also be observed in the polymeric matrix. The successive passes done to deposit the molted polymer left void spaces, as documented by Rodríguez [36]. Ekoi et al. [17] also reported a high porosity. Therefore, a high fiber content value may reduce the composite’s strength because of the higher voids between matrix and fiber [9]. In this study, the fiber content was fixed, but it is a parameter to consider when designing a composite part. Figure 14 also shows the reinforcing fibers on each side of the Onyx^®^ matrix. An enormous difference in the fracture shape is observed, the reinforcing fiber is thinner, and its fracture shape is slipped.

Figure 15a shows a fiber pullout from the matrix in an axial static test; mechanism reported in literature [17,23,30]. While Figure 15b shows fiber tearing because of the alternative bending fatigue test stress, in Figure 15a we can conclude that applied load is transmitted to the matrix, but the fibers are the ones withstanding the load. Such arrangement creates a strain gradient that pulls the fiber out of the matrix when it reaches a limit. Figure 15b shows the fiber for an alternative bending sample. We can observe the failed surfaces revealed fiber breakage mechanism, fiber tearing, and fiber buckling. Such mechanisms were summarized by Awaja [30].

## 6. Conclusions

The alternative bending fatigue behavior of a polymeric matrix composite material reinforced with continuous Kevlar fiber and printed by the novel Fused Filament Fabrication technique was analyzed. For this, the geometry of the specimens was established according to the ASTM D6272-17 standard. Static testing of said specimens gave a flexural modulus of elasticity of 4.73 GPa and flexural strength of 110 MPa.

The experimental stress versus the number of cycles (*S*–*N*) curve was obtained for the composite material. Alternative bending tests determined the curve according to ASTM D7774-17. From these tests, the failure cycles of the specimens were found, 15, 248, 460, and 711 cycles, corresponding to 92.3, 88.5, 86.2, and 84.7% of the maximum displacement, respectively. Finally, the composite material stiffness degradation was evidenced by the application of cyclic loads.

Numerical models of the manufactured specimens were created, and the static and fatigue behavior of the material was simulated using the ANSYS composite module. Results were obtained for mechanical properties such as modulus of elasticity and flexural strength, 4.48 GPa and 113 MPa, respectively. In the same way, numerically simulated cycles were very close to those obtained experimentally. That is 20, 250, 462, and 755 failure cycles for 92.3, 88.5, 86.2, and 84.7% of the maximum displacement, respectively. Thus, these numerical results were validated against experimental tests. Consequently, the computational model was validated using bulk mechanical properties instead of a more complex model that would require mechanical fatigue properties for each component of the composite material.

Inspection of the failed surfaces revealed the mechanisms of fiber breakage, matrix cracking, and matrix porosity for the static tests, whereas for alternative tests fiber tearing, fiber buckling, matrix cracking, and matrix porosity. The matrix porosity is seen as empty spaces between adjacent polymer layers. These types of failure mechanisms agree with the literature. Matrix cracking started at the site of maximum normal stress. No evidence of interlayer slipping or printing defects, other than commonly reported porosity, were observed.

## Figures and Tables

**Figure 1 polymers-14-03586-f001:**
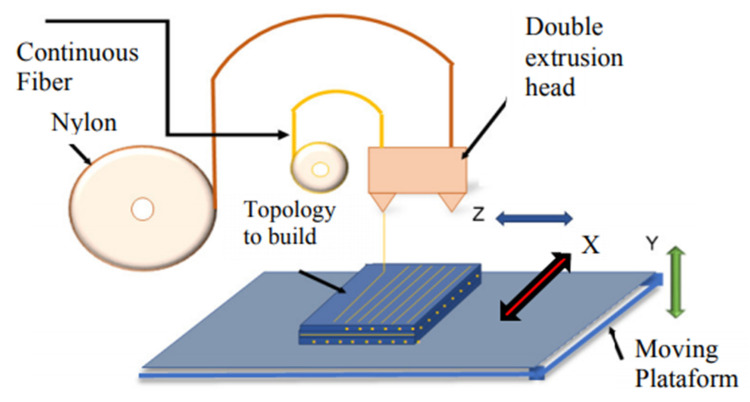
Schematic of the 3D printing process on the Markforged Two printer. Adapted from [2].

**Figure 2 polymers-14-03586-f002:**
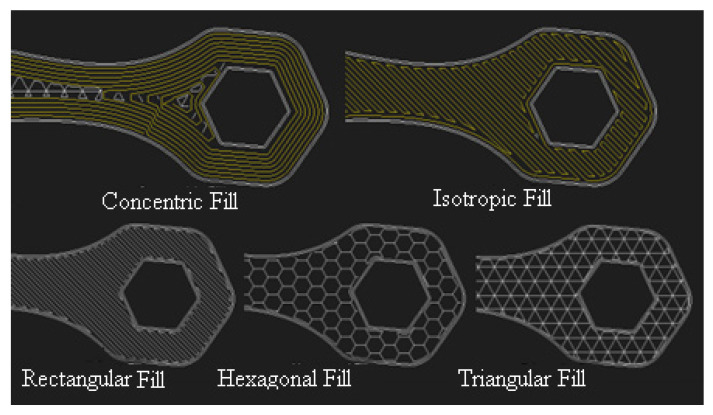
Schematic of fiber layout options and fill patterns in Eiger^®^ software, [9].

**Figure 3 polymers-14-03586-f003:**
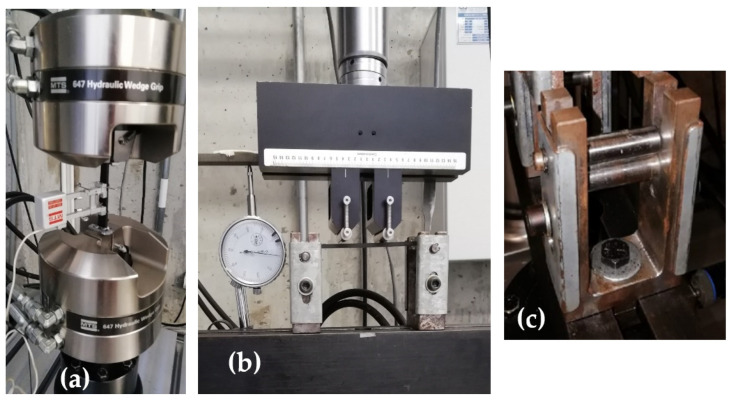
Specimens set up in the universal testing machine MTS 370.02, (**a**) tension test with axial extensometer MTS 632.12f, (**b**) static flexural test with a dial gauge, (**c**) side view of in-house built grips showing bidirectional support.

**Figure 4 polymers-14-03586-f004:**
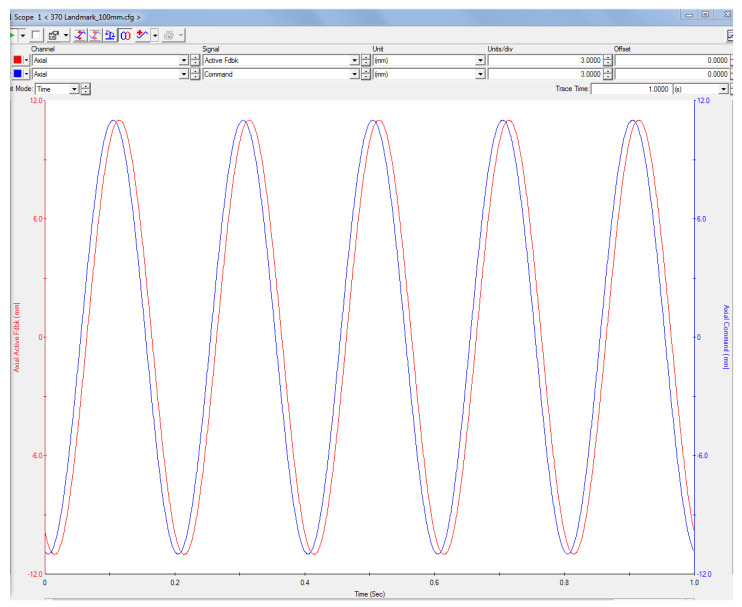
Screenshot of command and feedback signals for the MTS hydraulic machine.

**Figure 5 polymers-14-03586-f005:**
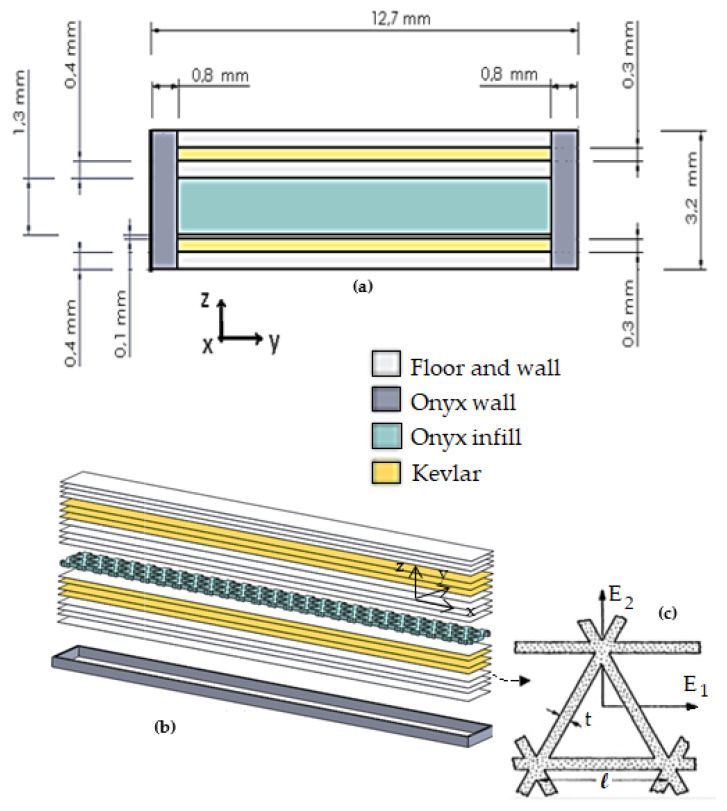
CAD layers in the detailed model, (**a**) cross-section, (**b**) organization of the material layers, (**c**) arrangement of the triangular fill.

**Figure 6 polymers-14-03586-f006:**
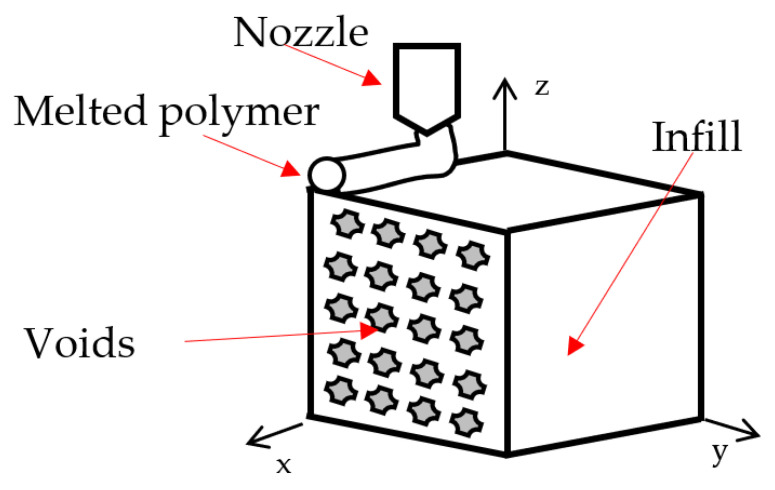
Representation of porosity level for the layers printed by FFF.

**Figure 7 polymers-14-03586-f007:**
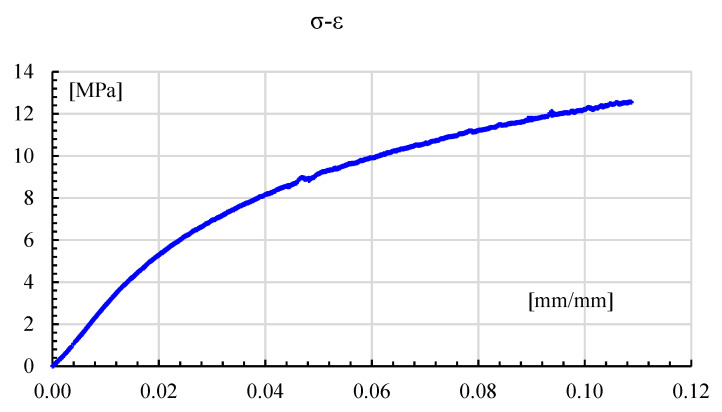
Stress-strain plot in axial tension for a non-reinforced Onyx^®^ sample.

**Figure 8 polymers-14-03586-f008:**
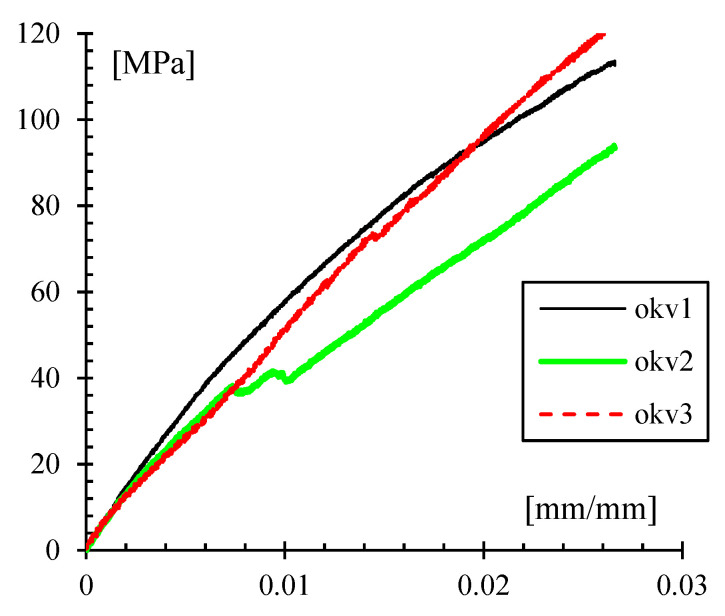
Stress-strain plot in four-point bending for Onyx^®^ reinforced with Kevlar.

**Figure 9 polymers-14-03586-f009:**
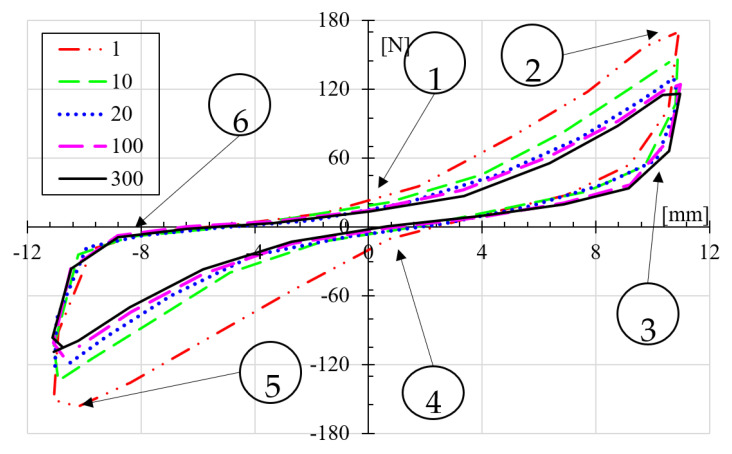
Exemplary load-displacement loops for a specimen subjected to ± 11 mm for different number of cycles.

**Figure 10 polymers-14-03586-f010:**
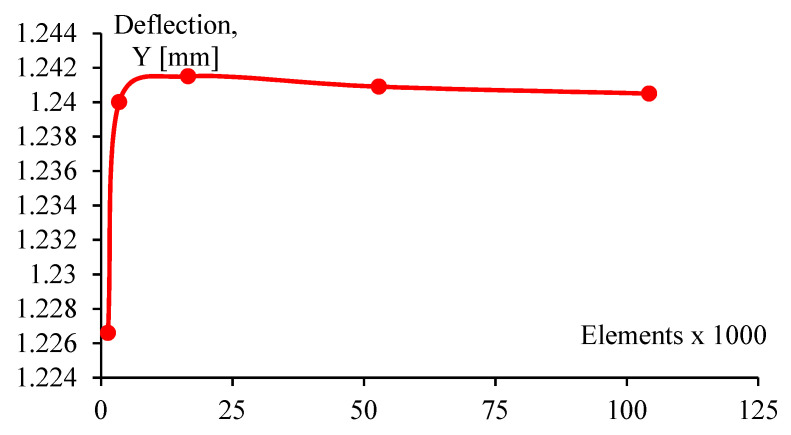
Exemplary convergence for vertical deflection values for the numerical simulation.

**Figure 11 polymers-14-03586-f011:**
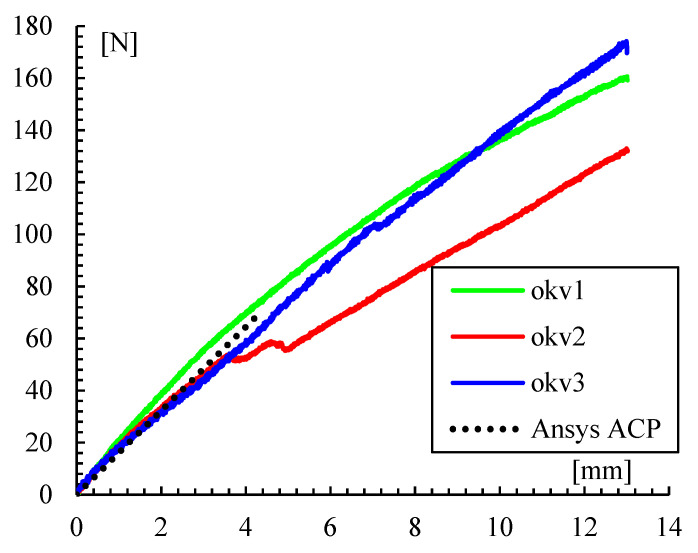
Load-displacement comparison of experimental results and the numerical model.

**Figure 12 polymers-14-03586-f012:**
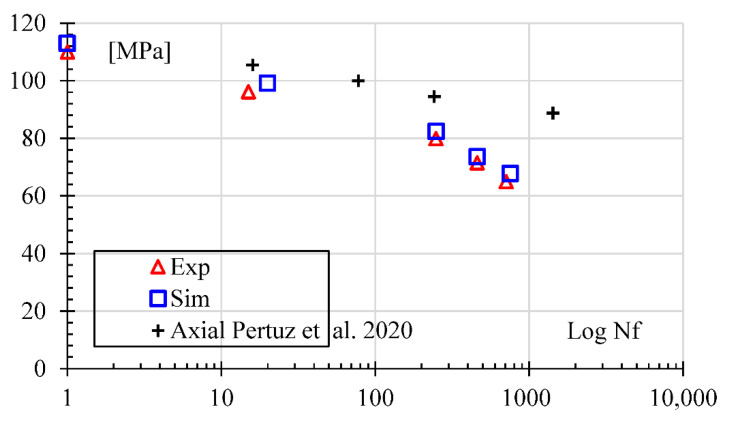
Stress-life curve for experimental and simulation results [15].

**Figure 13 polymers-14-03586-f013:**
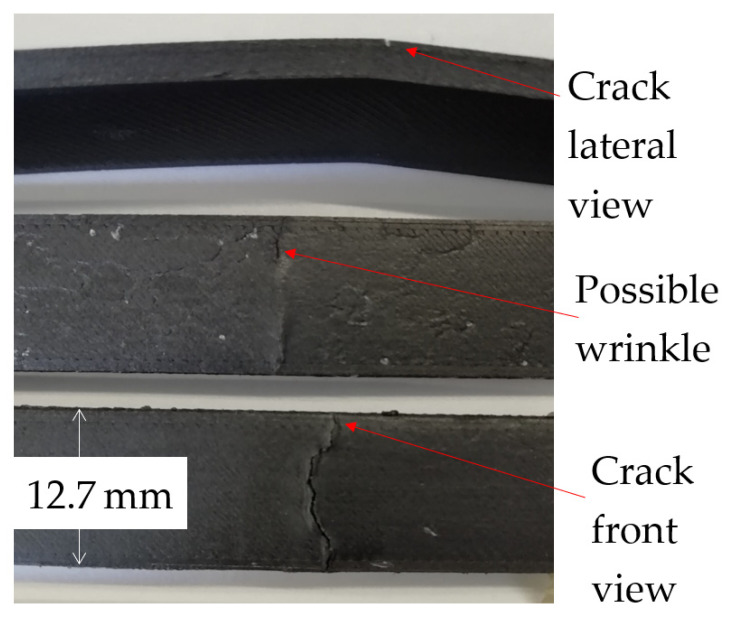
General view of the failure surface.

**Figure 14 polymers-14-03586-f014:**
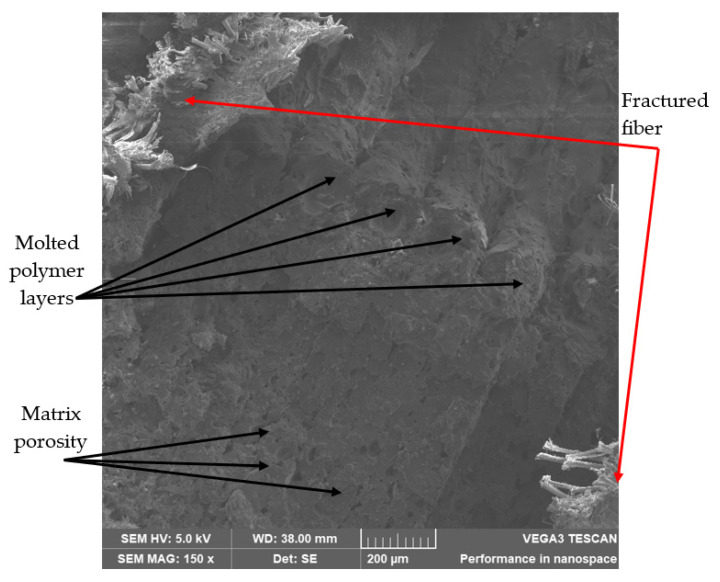
Onyx^®^ matrix showing porosity.

**Figure 15 polymers-14-03586-f015:**
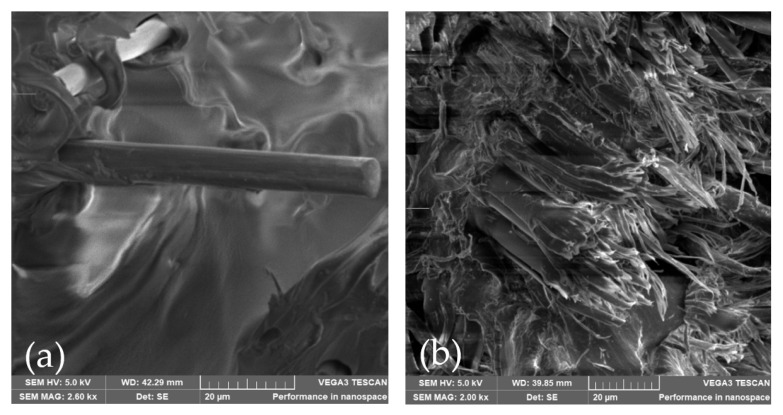
(**a**) close up of fiber pull out in axial static test, (**b**) fiber tearing in alternative bending.

**Table 1 polymers-14-03586-t001:** Individual mechanical properties as reported by Markforged [33].

Material	*E*_1_ Tension [GPa]	*σ*_1_ Tension [MPa]	*E*_1_ Bending [GPa]	*σ*_1_ Bending [MPa]	Relative Density
Onyx	2.4	40	3.0	71	1.1
Standard	ASTM D638	ASTM D638	ASTM D790	ASTM D790	NA
Kevlar fiber	27	610	26	240	1.2
Standard	ASTM D3039	ASTM D3039	ASTM D790	ASTM D790	NA

**Table 2 polymers-14-03586-t002:** Matrix printing parameters for the composite material, from [15,16].

Onyx^®^ Matrix	
Layer height	0.1 mm
Fill pattern type	Triangular
Matrix fill density	28%
Number of layers	4
Number of walls	2
Continuous fiber	Kevlar

**Table 3 polymers-14-03586-t003:** Properties of materials used for the static simulation.

Region	Young Modulus, [MPa]			Poisson Modulus			Shear Modulus, [MPa]		
	E_1_	E_2_	E_3_	v_12_	v_23_	v_13_	G_12_	G_23_	G_13_
Onyx^®^ (layers and wall)	1260	957.3	957.3	0.315	0.239	0.315	201.5	354.5	201.5
Kevlar	24873	2344	2344	0.378	0.503	0.378	682.7	559	682.7
Triangular filling	132	132	397.5	0.333	0.116	0.116	49.7	147.2	147.2

**Table 4 polymers-14-03586-t004:** Mechanical properties used for the fatigue simulation.

Onyx + Kevlar Simplified Model	
E_1_ (MPa)	4487
E_2_ (MPa)	895
E_3_ (MPa)	1011.5
v_12_	0.4535
v_23_	0.3701
v_13_	0.2566
G_12_ (MPa)	309
G_23_ (MPa)	287
G_13_ (MPa)	288

**Table 5 polymers-14-03586-t005:** Specimens tested in alternative bending.

Sample Group	Deflection (%)	Deflection, Y (mm)	*σ* (MPa)	*N_f_*
1	92.3	12	96.2	15
2	88.5	11.5	79.9	248
3	86.2	11.2	71.5	460
4	84.7	11.0	65.1	711

## Data Availability

Data available upon request.

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
