# Peer review of "Flexural Fatigue in a Polymer Matrix Composite Material Reinforced with Continuous Kevlar Fibers Fabricated by Additive Manufacturing"

_polymers, 2022, doi:10.3390/polym14173586_

Round 1
Reviewer 1 Report
In this manuscript, authors analyzed fatigue behavior of a polymer matrix composite material reinforced with continuous Kevlar fiber and printed by the Fused Filament Fabrication technique. Authors also proposed numerical models validated against experimental tests. I believe this work will be useful and/or provide reference for material fatigue study in industry or academy. In addition, the manuscript was written in a logical way for readers to follow: good introduction and background with sufficient references; informative illustration about experimental materials and methods; conclusions backed by results. I’d suggest acceptance after some minor revision or text editing (as shown below) :
• Page 1 – line 43, need to give full name for UTS;
• Cross-referencing of almost all figures or tables in the manuscripts were messed-up or missing “Error! Reference source not found..”
• Some styles of words in figures need correction (Fig 1, 13, 14, …)
• Page 8 – line 211 (a typo, Equation. (5).; supposed to be (6))
• Figure 9 – fatigue tests, what is the time or frequency for each cycle? Also, could author comment the time / frequency (fast vs slow) effect on the fatigue study?
Author Response
We would like to thank the anonymous reviewers for the valuable suggestions to improve the overall quality and understanding of the paper.
We have prepared a point-by-point response to the reviewer. Additionally, and for clarity, we have added a marked copy and a clean copy. We have included marked and clean versions as a general description of changes. The marked version contains changes made with MS-Changes tool. The clean version contains non-marked included and already deleted passages. We broke down the answers to the reviewer's concerns in different points as follows.
R1
In this manuscript, authors analyzed fatigue behavior of a polymer matrix composite material reinforced with continuous Kevlar fiber and printed by the Fused Filament Fabrication technique. Authors also proposed numerical models validated against experimental tests. I believe this work will be useful and/or provide reference for material fatigue study in industry or academy. In addition, the manuscript was written in a logical way for readers to follow: good introduction and background with sufficient references; informative illustration about experimental materials and methods; conclusions backed by results. I'd suggest acceptance after some minor revision or text editing (as shown below) :
- Page 1 – line 43, need to give full name for UTS;
We added the meaning of UTS
- Cross-referencing of almost all figures or tables in the manuscripts were messed-up or missing "Error! Reference source not found.."
It was corrected through all text. It seemed when the manuscript was changed to the journal format by the editorial office, those linked references were dropped.
- Some styles of words in figures need correction (Fig 1, 13, 14, …)
It was corrected in all images. It seemed when the manuscript was changed to the journal format by the editorial office, the original art was replaced by screenshots as *.jpge. So we changed the format to the original artwork. Moreover, we improved the quality of the pics, i. e. in Figure 13 we replaced the scale element at the bottom of the picture for the width of the specimen.
- Page 8 – line 211 (a typo, Equation. (5).; supposed to be (6))
It was changed
- Resp: Page 8 – line 211: added Equation before (8)
It was added
- Figure 9 – fatigue tests, what is the time or frequency for each cycle? Also, could author comment the time / frequency (fast vs slow) effect on the fatigue study?
Added the testing frequency. A note about such effect was also added, including a reference where its effect was discussed: ref [38] on the corrected version: H. Ma et al. 10.3390/polym14142772
We thank Reviewer #1 for the valuable suggestions
Reviewer 2 Report
The alternative bending fatigue behavior of a polymeric matrix composite material reinforced with continuous Kevlar fiber and printed by the novel melt deposition modeling technique was analyzed in this work. But I don't think this work is suitable for publication in polymers.
(1) Although the mechanical behavior of composites is analyzed and tested in detail, there is a lack of mechanical evidence and analysis.
(2) Although the paper adopts a novel melt deposition modeling technology, the overall innovation is not enough.
(3) Personal suggestions should be published in the Journal of mechanics.
(4) There are many grammatical errors and reference errors in the thesis. Some of the figures are not clear and standard.
Author Response
We would like to thank the anonymous reviewers for the valuable suggestions to improve the overall quality and understanding of the paper.
We have prepared a point-by-point response to the reviewer. Additionally, and for clarity, we have added a marked copy and a clean copy. We have included marked and clean versions as a general description of changes. The marked version contains changes made with MS-Changes tool. The clean version contains non-marked included and already deleted passages. We broke down the answers to the reviewer's concerns in different points as follows.
The alternative bending fatigue behavior of a polymeric matrix composite material reinforced with continuous Kevlar fiber and printed by the novel melt deposition modeling technique was analyzed in this work. But I don't think this work is suitable for publication in polymers.
(1) Although the mechanical behavior of composites is analyzed and tested in detail, there is a lack of mechanical evidence and analysis.
We believe that the use of cellular structure stiffness models, such as the ones provided by Gibson and Ashby, applied to 3D printed structures provide a solid mechanical background to predict and understand how these structures work.
(2) Although the paper adopts a novel melt deposition modeling technology, the overall innovation is not enough.
We believe that an adequate set of material parameters for models (such as Basquin) that can be used with accepted mechanical design methods is the key to popularizing the adoption of this kind of materials in mechanical designs. See ref [38] H. Ma et al. 10.3390/polym14142772 or ref [9] 10.1016/j.compositesb.2021.108657 where the same approach has been used or suggested.
(3) Personal suggestions should be published in the Journal of mechanics.
We do not understand this observation.
(4) There are many grammatical errors and reference errors in the thesis. Some of the figures are not clear and standard.
Thank you for pointing those out. We had the manuscript checked by Grammarly and by an American English native speaker. We believe grammatical errors and typos have been addressed.
We thank Reviewer #2 for the valuable suggestions
Reviewer 3 Report
Dear Authors,
The article presents an interesting research topic, however, it requires many changes, especially changes related to the quality of the article, improvement of the quality of photos, descriptions of figures and tables. I believe that the article should be carefully checked for errors and many changes, the key changes are presented below:
1. In the introduction, it is worth considering the analysis of rheological properties and thin-walled models, which will certainly improve the quality of the introduction, I recommended two articles: - Rheological Properties of Polyamide PA 2200 in SLS Technology - DOI10.17559 / TV-20190225122204 and second one - A Comparative Study of the Mechanical Properties of FDM 3D Prints Made of PLA and Carbon Fiber-Reinforced PLA for Thin-Walled Applications - DOI10.3390 / ma14227062
2. I think that the research in the second point, the manufacturing process should be described in much more detail, including all the key parameters.
3. In point 3 it would be worth mentioning the chemical composition and more properties.
4. In 3.2 should be explained in detail, some abbreviations were omitted.
5. Point 3.4 is very short, no photos, etc. here. it seems that this point should be developed here
6. Figure 9 should be described and explained in more detail.
7. Conclusions should be modified and extended as the summary of microscopy is poorly presented and the title describes the important ingredient being the introduction of fibers.
8. Point 5.3 should better describe in details especially these things which relate simulation and it condition etc.
Regards,
Reviewer
Author Response
We would like to thank the anonymous reviewers for the valuable suggestions to improve the overall quality and understanding of the paper.
We have prepared a point-by-point response to the reviewer. Additionally, and for clarity, we have added a marked copy and a clean copy. We have included marked and clean versions as a general description of changes. The marked version contains changes made with MS-Changes tool. The clean version contains non-marked included and already deleted passages. We broke down the answers to the reviewer's concerns in different points as follows.
The article presents an interesting research topic, however, it requires many changes, especially changes related to the quality of the article, improvement of the quality of photos, descriptions of figures and tables. I believe that the article should be carefully checked for errors and many changes, the key changes are presented below:
- In the introduction, it is worth considering the analysis of rheological properties and thin-walled models, which will certainly improve the quality of the introduction, I recommended two articles: - Rheological Properties of Polyamide PA 2200 in SLS Technology - DOI10.17559 / TV-20190225122204 and second one - A Comparative Study of the Mechanical Properties of FDM 3D Prints Made of PLA and Carbon Fiber-Reinforced PLA for Thin-Walled Applications - DOI10.3390 / ma14227062
We thank the reviewer for this valuable suggestion. A paragraph on page 2 was added describing the time dependency of PA6, and the suggested references were added.
- I think that the research in the second point, the manufacturing process should be described in much more detail, including all the key parameters.
We think the printing process, being a commercial solution, has been covered in other publications such as references [2, 14, 15, 16, 17, 24, 27, 29] and even the original patent reference [8] in the new version. So, adding a description would increase the paper length, which we believe is already long. Therefore, we added a sentence directing the reader to the original patent, reference [8] in the new version. But, on the other hand, and about the printing paraments, we would like to direct the reviewer to table 2 for the printing parameters description.
- In point 3 it would be worth mentioning the chemical composition and more properties.
We think the materials used, being a commercial solution, need no more description, plus we do not have that kind of information. Mechanical properties are listed in table 1. We added a line directing the reader to a reference that summarizes mechanical properties.
- In 3.2 should be explained in detail, some abbreviations were omitted.
Thank you for pointing those out. We added details about the temperature and testing frequency. We are not sure what the reviewer refers to the abbreviation. If it is about MTS, it is the company that makes the testing machine and software. We added its location to avoid future confusion; we also added the location of other companies as well.
- Point 3.4 is very short, no photos, etc. here. it seems that this point should be developed here
Section 3.4 is about Microscopy. The electron microscopy was standard using a commercial Vega 3 Tescan SEM. In order to keep the paper as short as possible and concentrate on new findings, we believe extra details about Microscopy can be consulted somewhere else by the interested reader.
- Figure 9 should be described and explained in more detail.
We added a note and compared results with what is reported in the literature to assess strength reduction.
- Conclusions should be modified and extended as the summary of Microscopy is poorly presented and the title describes the important ingredient being the introduction of fibers.
Thank you for the suggestion. The conclusion was extended to include findings about Microscopy as suggested.
- Point 5.3 should better describe in details especially these things which relate simulation and it condition etc.
The static simulation is described in the first paragraph of section 5.3, along with figures 10 and 11.
The fatigue simulation is described in the remaining paragraphs and plotted in figure 12. Fatigue Results at compared to the same material but under axial load. Finally, we would like to direct the reviewer to section 3.3 Numerical modeling, where is described how the static and fatigue simulations were performed.
We thank Reviewer #3 for the valuable suggestions
Round 2
Reviewer 2 Report
After the author's revision, I think this manuscript is ready for publication.
Reviewer 3 Report
Dear Authors,
The article can be published in presented form.
Regards,
Reviewer